



**An Orphan Problem Looking for Adoption**
**Responding to Ocean Acidification Utilising Existing International Institutions**
Ellycia R. Harrould-Kolieb[1]
[1]Australian-German Climate and Energy College, University of Melbourne, Melbourne, 3000, Australia
*Correspondence to*: Ellycia R. Harrould-Kolieb (ellycia.harrould@unimelb.edu.au)



**Abstract**
Ocean acidification poses a substantial threat to the ocean, marine wildlife and the goods and services they
provide. As a result it presents a substantial regulatory challenge at the international, regional, national and sub-
national levels. In the international realms, ocean acidification is not currently addressed by any international
instrument or stand-alone agreement, nor does there appear to be any coherent framework for responding to the
issue. Despite this, there are a number of international institutions, including treaty bodies and specialised UN
agencies that have expressed an interest in ocean acidification and have begun to initiate an array of relevant
activities – a small number of which may be considered substantive activities, including rule-making and
implementation.
This paper is an effort to explore the existing international frameworks that are applicable to forming a response
to ocean acidification in an attempt to prevent worsening acidification and respond to impacts now and into the
future. Six policy domains are outlined that together comprise a comprehensive response to ocean acidification.
Each of these are then addressed with respect to what institutions are currently doing to respond to acidification
and what could be done in the future.
This paper finds that only three international institutions have initiated substantive policy-making in response to
ocean acidification with respect to the regulation of carbon capture and storage and the protection of species.
While these are important policy interventions, they are simply not enough to prevent worsening ocean
acidification or respond to the impacts resulting from increased acidity, even when coupled with policies, such
as regulation of carbon dioxide under the UNFCCC that have been implemented without reference to ocean
acidification. In order to fill the existing gaps, this paper proposes a series of, as yet unutilised mechanisms that
could be employed to enhance a response to ocean acidification.



## 1. Introduction

The problem of ocean acidification is a complex global issue, resulting primarily from the emission of anthropogenic carbon dioxide ($CO_2$) (Doney et al., 2009) and yet, is exacerbated by a myriad of local stressors (Cai et al., 2011;Hassellöv et al., 2013). Its impacts are present across many scales, from the microscopic, through to ecosystems and on to the global (Gattuso and Hansson, 2011). Its consequences are not limited by geography and are felt in and across national boundaries and in areas of the global commons. However, consequences are not experienced evenly, sometimes with those least responsible the most vulnerable. Further, ocean acidification has implications for biodiversity (Sutherland et al., 2009), economic stability (Narita et al., 2012) and sustainable development (Rockström et al., 2009) and its solutions are intimately tied with other complex global problems, such as climate change. Ocean acidification and its consequences are, therefore, pertinent to and present challenges for the work of a number of international institutions and yet, a response does not appear to fall neatly with the mandate of any. Thus, causing it to sit "within a very complex institutional landscape, at a rather cracked interface between the climate, biodiversity and oceans regimes" (Kim, 2012 p.257).

With no treaty or international instrument designed deliberately to address ocean acidification the issue is somewhat relegated to the 'twilight zone' with no single institution responsible for guiding a response. Despite this, there are a number of international institutions, including treaty bodies and specialised UN agencies that have expressed an interest in ocean acidification (UNGA, 2006;CCAMLR, 2009;UNFCCC, 2015b). However, much of this interest appears to be limited to calls of concern and knowledge production activities, with limited efforts to change legal frameworks or initiate implementation policies to integrate ocean acidification into existing structures (Billé et al., 2013).

Given this lack of substantive policy making, one is left asking what can be done to enhance the global governance of ocean acidification. One avenue that has been proposed is the creation of a comprehensive ocean acidification treaty, that would tackle all aspects of a response in one forum (Lamirande, 2011;Kim, 2012). However, such an effort seems unlikely at this time, with seemingly little support in the wider policy or academic communities. Thus, we are left attempting to fill the gaps by utilising existing international mechanisms to respond to ocean acidification.

This paper is, therefore, an effort to explore in more depth the existing international frameworks that are applicable to ocean acidification and can be utilised to take action. This paper proceeds by first exploring the problem of ocean acidification and its solutions. Six policy domains are proposed that need to be filled in order to prevent worsening acidification and address its impacts now and into the future. The discussion then turns to a review of activities initiated by the United Nations (UN) and UN affiliated bodies as well as international treaties deposited with the UN that have been implemented, at least in part, as a response to ocean acidification. This review reflects upon their capacity to fill all six policy domains. Substantial gaps are found, hence, non-acidification directed policies are accessed to see if it is possible that ocean acidification is being addressed without explicit intent to do so. Again the responses are found lacking. Thus, this paper turns to an exploration of existing mechanisms and institutions, which are not yet being applied to ocean acidification, to investigate how they can be utilised to further contribute to a response.



## 2. Responding to Ocean Acidification

An ocean acidification response has two main objectives: preventing acidification from worsening, while simultaneously addressing the impacts that have already occurred (or those that are yet to occur due to already released emissions). Billé et al. (2013) identified the main way to achieve these two goals. First and foremost is the need to limit carbon dioxide concentrations in the atmosphere. This can be achieved by the direct reduction of carbon dioxide emissions or the removal of carbon dioxide from the atmosphere. Reducing the local factors that cause acidification, including nutrient inputs, can also help to prevent worsening acidification. Strengthening ecosystem resilience to ocean acidification, adapting human activities in anticipation of, or reaction to, ocean acidification, and repairing damages when the ocean has already acidified by restoring degraded systems or reducing acidity using additives other than iron are all measures that can be used to address the impacts of rising ocean acidity.

This range of available responses can be grouped under six types of policy domains (as summarised in Table 2). *Mitigation* policies are those intended to lead to reductions in carbon dioxide emissions, while *non-CO$_2$ mitigation* policies are those aimed at reducing non-CO$_2$ emissions that contribute to OA as well as efforts to reduce or remove other local exacerbating factors, such as run-off. *Adaptation and Protection* measures are policies aimed at enhancing resilience in human and ecological communities to enable them to better withstand the impacts of ocean acidification. These are grouped together as the terms adaptation and protection appear to be used interchangeably within a number of policy settings and can refer to either human or ecological communities. *Restoration* policies are those intended to facilitate the repairing and rebuilding of ecological communities harmed by ocean acidification, while *reparation* policies are those implemented to assist human communities that have suffered damage or loss. Finally, those policies aimed at manipulating oceanic or atmospheric properties to address ocean acidification, whether they be mitigation, restoration or other type policies, are termed *geoengineering*, such as addition of additives to increase alkalinity.

**Table 1: Interventions for Preventing Worsening OA**

| POLICY DOMAIN | OBJECTIVE | EXAMPLES |
|---|---|---|
| CO$_2$ Mitigation | Reducing the primary global driver of ocean acidification | Introduce renewables<br>Increase efficiency<br>Land use changes |
| Non-CO$_2$ Mitigation | Reducing the local factors that exacerbate ocean acidification | Reduce runoff<br>Reduce Non-CO$_2$ emissions |
| Adaptation and Protection | Building or maintaining resilience in order to assist human communities and ecological systems to overcome, absorb, return from or adjust to change | Establish marine protected areas<br>Reduce non-OA stressors<br>Alter species distribution<br>Protect ecosystem services<br>Identify alternative sources of income<br>Alter commercial/industrial practices |
| Restoration | Repairing ecological communities after damage has occurred | Replant vegetation<br>Reseed coral reefs<br>Reintroduce species |
| Reparations | Ameliorating damage that has occurred within human communities | Establish reparation funds |
| Geoengineering | Altering the physical properties of the ocean or atmosphere to prevent further acidification or reduce acidity | Sequester CO$_2$<br>Remove CO$_2$ from atmosphere<br>Increase alkalinity of water |

This suite of options for tackling ocean acidification begins with the reduction of carbon dioxide in the atmosphere. It is the primary solution to halting further increases in acidity as it is largely these emissions that





will determine the trajectory of the acidity of the global ocean in the near future (Caldeira and Wickett, 2003).
This can be achieved, for example, via the introduction of renewable energy and the removal of fossil fuels,
increases in efficiency of energy production, changes in land use and the capture and storage of carbon dioxide.
In addition to carbon dioxide, local factors can exacerbate or alleviate this global trend and therefore for some
locations the removal or reduction of these factors can be an important way of protecting discreet geographic
areas, for example, limiting run-off (Cai et al., 2011) and reducing emissions, such as sulphur and nitrogen
emissions from shipping (Doney et al., 2007).
While mitigation is the only way to prevent long term increases in acidity, negative effects are already occurring
(De'ath et al., 2009;Bednaršek et al., 2014) and will continue to occur due to the locked in impacts from already
released emissions (Joos et al., 2011). Thus, efforts to build and maintain resilience in order to assist human
communities and ecological systems withstand, absorb, or adjust to these changes are also required. These types
of policies can be targeted at protecting ecological communities, for example by establishing marine protected
areas, clearance of invasive species and the removal of other anthropogenic stressors, such as pollution (Billé et
al., 2013). Alternatively, policies can be targeted at ensuring human communities have the potential to adapt to
changes, for example, by switching fisheries targets to less vulnerable species (Ekstrom et al., 2015) or
establishing monitoring systems to allow commercial enterprises to respond appropriately to changing pH
levels, as has occurred in Washington State oyster hatcheries (Barton et al., 2015).
It is also possible that efforts to reduce emissions, protect ecological systems and enhance the adaptive capacity
of human communities may simply not be enough in some cases. As a result it is also important to consider
whether human communities may be entitled to reparations for damages and loss experienced due to the impacts
of ocean acidification. In addition, efforts will be needed to determine whether degraded ecosystems can and
should be restored, and if so, via what methods. For example, the reintroduction of species, reseeding coral
reefs, or increasing ocean pH via introduction of various additives (Rau et al., 2012).

## 3. Current Responses to Ocean Acidification

These six domains offer a typology for examining how the international community, via the UN and its
affiliated institutions, is either preventing ocean acidification from worsening or responding to impacts that have
already occurred. This research revealed that only three institutions, the London Convention and Protocol
(LC&P), the OSPAR Convention, and the Convention on Biological Diversity (CBD), have implemented any of
these types of policies with explicit consideration of ocean acidification.[1]
The Convention on the Prevention of Marine Pollution by Dumping of Wastes and Other Matter (LC, 1972),
known as 'The London Convention' or the 'dumping Convention' was designed to prevent pollution of the
ocean from the dumping of waste from vessels, aircrafts and platforms. The Convention functions by providing
a banned list of substances that cannot be disposed of in the marine environment. The London Protocol (1996),
designed to replace the Convention is more precautionary and provides a 'reverse list' naming substances that
may be considered for dumping, while prohibiting all others (Annex 1). The Convention on Biological Diversity

---

[1] A number of other activities were found to have been initiated in response to ocean acidification, including calls of concern, calls for action and knowledge production, all of which are important and contribute to the larger discourse around ocean acidification and will (hopefully) lead to further policy making. However, these activities are beyond the scope of this particular discussion as they are not classified as policies designed to directly prevent further acidification or address the impacts that have already occurred.



(1992a) was designed to conserve biological diversity, as well as its sustainable and equitable use. The
Convention provides a framework for national action via agreed upon goals and guidelines, without putting in
place many binding obligations, beyond the obligation to address the issues covered by the Convention
(Secretariat, 2001).Both the LC&P and the CBD are conventions with global scope and both enjoy fairly high
participation rates (with 87, 48 and 196 members respectively). The Convention for the Protection of the Marine
Environment of the North-East Atlantic, or the "OSPAR Convention" (1992) is a regional convention focused
on the protection of the North-East Atlantic and has 18 contracting parties. The objectives of the OSPAR
Convention are implemented via the adoption of decisions (which are legally binding) and recommendations
and other agreements that guide the activities of its members.

### 3.1 Geoengineering

As early as 2004, ocean acidification began to appear in discussions around the possible placement of carbon
dioxide in the OSPAR maritime area as a way of addressing climate change. The potential detrimental effects of
ocean acidification due to increasing anthropogenic carbon dioxide were highlighted and a review of existing
knowledge was commissioned (OSPAR, 2005). This resulted in the publication of a technical report, *Effects on*
*the marine environment of ocean acidification resulting from elevated levels of $CO_2$ in the atmosphere*, which
provided an overview of ecosystem sensitivity to carbon dioxide exposure (Haugan et al., 2006).
In 2007, the OSPAR Commission (the decision-making body of the Convention) formally expressed concern
over the 'implications for the marine environment of climate change and ocean acidification due to elevated
concentrations of carbon dioxide in the atmosphere' (OSPAR, 2007b, p. 1). The Commission further recognised
that the storage of carbon dioxide in geological formations could act as part of a portfolio of measures for
mitigating these impacts (OSPAR, 2007a). OSPAR adopted a *Decision* to ensure environmentally safe storage
of carbon dioxide streams in geological formations, while legally ruling prohibiting the placement of carbon
dioxide streams in the water column or on the seabed, due to the likelihood of resulting harm to living resources
and marine ecosystems (OSPAR, 2007b).
Echoing the discussions taking place within the OSPAR regime, the Consultative Meeting of Contracting
Parties to the London Convention acknowledged, in 2005, that carbon dioxide posed a direct threat to the marine
environment and was responsible for causing ocean acidification. It was also acknowledged that carbon dioxide
sequestration and storage, which had effectively been banned until this point, could bring about benefits to the
oceans in terms of reducing ocean acidification and climate change. Furthermore, it was agreed that the act of
carbon sequestration and its implications for the marine environment came under the purview of the LC&P
(IMO, 2005).As a result, in 2006 an amendment was made to Annex 1of the Protocol (the 'reverse list') that
allowed for the consideration of dumping of 'carbon dioxide streams from carbon dioxide capture processes for
sequestration' (IMO, 2006, p.3). It was decided that carbon dioxide may only be considered for dumping if
'disposal is into a sub-seabed geological formation'(IMO, 2006, p.3), thereby effectively maintaining a
prohibition on its disposal in the water column or on the sea floor.

### 3.2 Protection and Adaptation

In 2010, the OSPAR Commission *agreed* to 'monitor and assess the nature, rate and extent of the effects of
climate change and ocean acidification on the marine environment and consider appropriate ways of responding
to those developments' and that '[c]onsiderations of the impacts of climate change and ocean acidification, as



well as the need for adaptation and mitigation, will be integrated in all aspects of the work'(OSPAR, 2010, p.3).
Significantly, the Commission also *agreed* that it would strengthen the OSPAR network of marine protected
areas in recognition of their role in 'maintenance of ecosystem integrity and resilience against human activities
and impacts of climate change and ocean acidification' (OSPAR, 2010, p.5).
Ocean acidification first began to appear in discussions within the CBD in 2008 when it was considered at the
Subsidiary Body on Scientific, Technical and Technological Advice (SBSTTA)(SBSTTA, 2008). It then made
its way on to the agenda of the 9[th] Convention of the Parties (COP), at which it was requested that the Executive
Secretary, in conjunction with others, prepare a synthesis report of available scientific information pertaining to
ocean acidification (CBD, 2008). The resulting report, *Scientific Synthesis of the Impacts of Ocean Acidification*
*on Marine Biodiversity* (Secretariat, 2009), was considered at the following SBSTTA meeting. At which it was
recommended that the COP adopt a decision expressing serious concern about increasing ocean acidification
and the potential threat to biodiversity and ecosystems and the consequent impacts on the services they provide
(SBSSTA, 2010). SBSTTA also recommended that the COP request the Executive Secretary to, in conjunction
with other relevant organisations and scientific groups, develop a series of expert review processes to monitor
and assess the impacts of ocean acidification and widely disseminate the result to raise awareness both within
the CBD and without. SBSTTA also suggested that given the relationship between atmospheric carbon dioxide
concentration and ocean acidification the COP request the Executive Secretary to transmit the findings to the
Secretariat of the UNFCCC (SBSSTA, 2010). All of these recommendations were accepted at the 10[th] COP, at
which the COP expressed 'its serious concern that increasing ocean acidification, as a direct consequence of
increased carbon dioxide concentration in the atmosphere, reduces the availability of carbonate minerals in
seawater…'(CBD, 2010b p.12)
The CBD COP also adopted a list of Ecologically or Biologically Significant Marine Areas (ESBAs) and
encouraged their conservation and sustainable use. These areas were identified as serving an important purpose
in supporting the healthy functioning of the ocean and included the *Western South Pacific high aragonite*
*saturation state zone.* An area identified as having the highest aragonite saturation state in the ocean today and,
therefore, the last to fall below critical thresholds with increasing acidification (CBD, 2012). This area,
therefore, may be the slowest to be impacted by ocean acidification and potentially the fastest to recover.
Significantly, the COP also set out a revised and updated strategic plan for biodiversity for 2011-2020, which
included establishing new biodiversity targets, the "Aichi Targets"(CBD, 2010c). The Aichi Targets set out a
series of goals aimed at halting the loss of biodiversity by 2020. Target 10 recommends that 'the multiple
anthropogenic pressures on coral reefs, and other vulnerable ecosystems impacted by climate change or ocean
acidification are minimized, so as to maintain their integrity and functioning' (CBD, 2010c, p.119). The rational
provided for this target is that the reduction of stressors affecting ecosystems will help to make them less
vulnerable to the impacts of acidification over the short to medium-term, thus, providing more time to address
acidification over the longer-term. 'Ultimately the aim of this target is to provide ecosystems with the greatest
probability of maintaining their integrity and functioning despite the effects of climate change and/or ocean
acidification' (CBD, 2013, p.1). Pollution control, reducing over-exploitation and harvesting, eradication of
invasive species are all activities offered as ways to reach this target. While remaining fairly vague, this is



significant as it is the first target set by any international institution with a timeframe for responding to ocean
acidification.
In response to this target, SBSTTA suggested a series of practical responses available to Parties to meet Target
10 and help reduce threats from ocean acidification. With regards to mitigation, Parties were encouraged to
work towards emission reductions of carbon dioxide and to participate in the UNFCCC, IPCC and other related
processes. These are relatively vague and aspirational and it appears that mitigation activities have largely been
deferred to other bodies, such as the UNFCCC, that are deemed more relevant to the task. However, the
guidance offered for maintaining and restoring ecosystem resilience is far more detailed and includes specific
activities that governing bodies can implement, including effectively managing coastal runoff, limiting the
impacts of unstainable fishing practices and the reduction of local pollutants (SBSSTA, 2012).

### 219 3.3 Substantial Gaps in the Response to Ocean Acidification

The substantive activities of the CBD, OSPAR Convention and the LC&P, as summarised in Table 2, are useful
first steps in crafting an international response to ocean acidification. However, these policies by themselves are
unable to prevent worsening acidification. This is in large part because these activities focus on the protection of
species and ecosystems and the regulation of geoengineering efforts and do not tackle the root cause of ocean
acidification – rising carbon dioxide emissions. Activities focused on alleviating local pressures, protecting and
restoring ecosystems and helping human communities to adapt and respond are critical to ensuring positive
outcomes in the face of ocean acidification. However, the success of interventions designed to alleviate the
pressure of ocean acidification greatly declines with increasing emissions (Gattuso et al., 2015). Thus, these
options are only viable when coupled with substantive action to reduce carbon dioxide emissions. Without
measures to reduce carbon dioxide, non-$CO_2$ interventions become more costly and less effective and are only
capable of delaying the worsening impacts of ocean acidification for a short period of time (Kennedy et al.,
2013). As a result, non-$CO_2$ mitigation, protection, adaptation, restoration and reparation efforts, while
important, remain largely ancillary to $CO_2$ reduction efforts and should only be viewed as effective when
coupled with $CO_2$ emission reductions.
**Table 2: Policies Initiated with Explicit Intent to Respond to Ocean Acidification**

| INTERVENTION | OSPAR Convention | LC&P | CBD |
|---|---|---|---|
| CO$_2$ Mitigation | | | |
| Non-CO$_2$ Mitigation | | | Recommended minimization of anthropogenic pressures on ecosystems impacted by OA |
| Adaptation and Protection | Agreed to strengthen the OSPAR network of marine protected areas | | Identified Western South Pacific high aragonite saturation state zone for protection |
| Restoration | | | |
| Reparations | | | |
| Geoengineering | Prohibited the disposal of carbon dioxide on or above the sea floor | Allowed for the sequestration of CO$_2$ in sub-seabed geological formations. | |





## 4. Co-Benefits of Non-Ocean Acidification Directed Policies

It is evident that there are substantial gaps in the current governance of ocean acidification, especially in the domains of carbon dioxide mitigation, restoration and reparations. Even within those domains with existent policies there is room for additional efforts to create a more robust response to acidification. However, it is possible that efforts already exist that have been initiated without consideration of ocean acidification, that may actually be deemed relevant to its response.

### 4.1 $CO_2$ Mitigation

Most significant is the work being undertaken within the United Nations Framework Convention on Climate Change (UNFCCC)(1992a) to regulate emissions of carbon dioxide. As the main international institution working to regulate carbon dioxide emissions, it is this institution that has the largest potential to determine future levels of ocean acidity. To date, there has been little activity on behalf of the COP to consider ocean acidification in discussions of targets and timelines for emission reductions. Nevertheless, rapid decarbonisation in order to address climate change would also address ocean acidification. In the most recent Paris Agreement, Parties agreed to hold 'the increase in the global average temperature to well below 2°C above pre-industrial levels and pursuing efforts to limit the temperature increase to 1.5°C above pre-industrial levels'(UNFCCC, 2015a, p.3).

This agreement, paves the way for large-scale emission reductions, resulting in decarbonisation, thereby preventing future acidification. However, the agreement also leaves room for less ambitious action, including surpassing a 1.5°C rise in global temperatures, delaying a reduction to net zero emissions by as late of the end of the century and utilising technologies to remove substantial amounts of carbon dioxide from the atmosphere later in the century. Such scenarios would allow for continued high emissions in the short-term and rapid reductions at a later time, which would result in worsening acidification and irreversible impacts in the near future (Mathesius et al., 2015). The UNFCCC Expert Review suggested that there is a high likelihood of a meaningful difference in impacts resulting from global temperature increases of 1.5 or 2°C. At 1.5°C risk from acidification is likely to be on the verge of high risk, whereas at 2°C the risk would already be high. In addition, an overshoot of the target followed by a rapid reduction in emissions would likely result in impacts from acidification, irreversible for tens of thousands of years due to slow ocean processes (UNFCCC, 2015b). Thus, it is difficult to conclude that the Paris Agreement, unless implemented in its most stringent form, is strong enough to prevent a worsening of acidification into the future. As a result, there is still a need to work towards stronger targets and timelines with consideration of ocean acidification.

There has been additional work to reduce carbon dioxide emissions undertaken within the International Convention for the Prevention of Pollution from Ships (MARPOL)(1973). MARPOL has taken steps to regulate carbon dioxide emissions from the shipping industry, which account for approximately 2.2 percent of global emissions (MARPOL, 2017), via the introduction of operational and technical measures (MEPEC, 2011). While not implemented with reference to ocean acidification, nor comprising a big enough reduction in global emissions to prevent further acidification, this is significant as it is the first mandatory regime for regulating the emissions of a global industry. Such measures will aid in attempts to reduce global emissions and paves the way for other industry specific regulations to occur within other institutions.



### 4.2 Non-CO$_2$ Mitigation

MARPOL has also been instrumental in setting limits on the emissions of sulphur and nitrogen and other
pollutants from ships. Again, these regulations have been put in place to reduce air pollution and not as an
attempt to respond to ocean acidification. Although a 2010 submission by the United States proposing areas to
be designated as Sulphur Emission Control Areas noted that sulphur and nitrogen deposition from ships causes
local acidification of marine waters  (MPEC, 2010).

### 4.3 Geoengineering

Along with efforts to reduce carbon dioxide and non-CO$_2$ emissions a number of efforts have been initiated to
regulate marine geoengineering. This is significant as some of these activities are thought likely to exacerbate
ocean acidification (Cao and Caldeira, 2010). Concerns have been raised within the CBD and LC&P over the
effectiveness and possible negative impacts on the marine environment of iron fertilization (with no mention of
ocean acidification). In 2008, the CBD COP requested Parties and urged other governments to 'ensure that
ocean fertilization activities do not take place until there is an adequate scientific basis on which to justify such
activities'(CBD, 2008). Noting this decision, the LC&P placed a moratorium on all ocean fertilization activities
(excluding those conducted for legitimate scientific research purposes)(LC&P, 2008).Further, in 2010 the CBD
COP adopted a decision that invited Parties and other Governments to ensure 'that no climate –related geo-
engineering activities that may affect biodiversity take place, until there is an adequate scientific basis on which
to justify such activities and appropriate consideration of the risks for the environment and biodiversity and
associated social, economic and cultural impacts' (CBD, 2010a, p.5). While these steps are useful in regulating
geoengineering, it is possible that such efforts will still go ahead, thus there is a need to consider their impacts,
positive, negative, or benign, for ocean acidification prior to their deployment.

### 4.4 Protection, Adaptation and Restoration

Also of relevance to ocean acidification are the multitude of conservation measures that have been implemented
under various institutions. Listing all is simply beyond the scope of this paper; however, efforts include
establishing marine protected areas, limiting fishery quotas, and restoration of local habitats. These policies may
play a role in boosting resiliency and protecting biodiversity from increasing acidity, as well as restoring
impacted systems and aiding human communities in adapting to changing conditions. However, conservation
measures implemented without consideration of the trajectory and impacts of ocean acidification may allow
activities that will exacerbate ocean acidification. In addition, they may simply not be constructed or
implemented in a way that helps human and ecological communities to overcome the impacts of ocean
acidification. For instance, it is not enough to have areas designated as protected, it is recommended that they be
specifically located to avoid hotspots of acidification  (Hofmann et al., 2011;Kelly et al., 2011), while
simultaneously placed to act as refugia, either by preserving areas that are likely to acidify at a slower rate or by
protecting populations that exhibit high levels of genetic diversity and natural resilience (Billé et al., 2013).
Thus, if institutions wish to protect and restore ecological communities and aid the human communities
dependent upon them, conservation measures need to be designed with ocean acidification in mind.

### 4.5 Gaps Still Exist

It is evident that there are a number of existing policies, initiated without consideration of ocean acidification,
which are able to help lessen worsening acidification and address impacts (See Table 3 for a summary). Efforts



to reduce carbon dioxide emissions within the UNFCCC have, to date, not considered ocean acidification and
thus, are not strong enough to prevent increasing acidification in the future. Other mitigation policies, including
those within MARPOL provide positive steps to reduce industry wide emissions of carbon dioxide, however are
not broad enough to capture a large enough segment of emissions so as to prevent future acidification. In
addition, general conservation measures are likely to have a positive effect on ecosystems in the face of rising
acidity. However, without specific intent to address acidification it is possible that such measures could miss
important opportunities with regards to protecting ecological systems. Further, few legally binding restrictions
have been placed on deployment of geoengineering efforts and thus, may be used in the future. It is important to
understand how such efforts could interact with ocean acidification and ensure that the possible negative
impacts are considered prior to their deployment.
**Table 1: Existing Policies that Form a Response Ocean Acidification**

| INTERVENTION | OSPAR Convention | LC&P | CBD | UNFCCC | MARPOL |
|---|---|---|---|---|---|
| CO₂ Mitigation | | | | Agreed to rapid reductions of GHG emissions | Operational and technical measures to reduce ship emissions |
| Non-CO₂ Mitigation | | | Recommended minimization of anthropogenic pressures on ecosystems impacted by OA | | SOx and NOx regulations |
| Adaptation and Protection | Agreed to strengthen the OSPAR network of marine protected areas | | Identified Western South Pacific high aragonite saturation state zone for protection | | |
| Restoration | | | | | |
| Reparations | | | | | |
| Geoengineering | Prohibited the disposal of carbon dioxide on or above the sea floor | -Allowed for the sequestration of CO₂ in sub-seabed geological formations. -Moratorium on iron fertilization and other forms of marine geoengineering | Requested ocean fertilization activities do not occur | | |


It is appears that while there are a series of policies currently forming an international response to ocean
acidification, they are simply not enough, even when coupled with non-acidification directed efforts, to prevent
the worsening of ocean acidification or address its impacts. Thus, this paper will now turn to a discussion of
existing policies within international institutions that are not currently being utilised to respond to ocean
acidification that could be employed to enhance efforts to prevent worsening of ocean acidification and respond
to impacts as they occur.



## 5. Utilising Existing International Instruments

### 5.1 CO$_2$ Mitigation

With regards to the mitigation of carbon dioxide emissions, the UNFCCC remains the venue in which the international community has come together to regulate emissions. As discussed above the Paris Agreement has established a long-term goal for emission reductions that provides a pathway for avoiding unacceptable risks associated with both ocean acidification and climate change. However, the leniencies built into the agreement mean that this is not guaranteed. Thus, there is still a need for the broader incorporation of ocean acidification into discussions within the UNFCCC (Harrould-Kolieb, 2016). This could be worked into the periodic reviews that will take place in regards to strengthening the long-term goal and the timeline for meeting this goal. Some scholars, however, suggest that this is unlikely to occur due to structural limitations of the UNFCCC mandate that effectively prevents a more meaningful consideration of ocean acidification within the workings of the Convention (Baird et al., 2009;Kim, 2012). However, these are narrow readings of the Convention and do not take into account its progressive nature with regards to the incorporation of developing science (Harrould-Kolieb, 2016). Thus, it is possible and critical for ocean acidification to be considered alongside climate change when setting targets, timelines and methods for emission reductions within the UNFCCC.

It is worth noting that a number of other fora, including the United Nations Convention on the Law of the Sea (UNCLOS)(1982) and the United Nations Fish Stocks Agreement (UNFSA)(1995) have been proposed as avenues for limiting carbon dioxide emissions, primarily due to their obligations to protect the marine environment through the regulation of pollutants. These institutions are seen as particularly attractive as they both have binding dispute resolution mechanisms in place that could, it is proposed, essentially be used to compel states to reduce their carbon dioxide emissions (Boyle, 2012;Burns, 2006). However, these institutions are unlikely to be utilised, primarily because of the significant duplication of efforts already being pursued within the UNFCCC to reduce emissions (Boyle, 2012). These institutions are also less widely subscribed to than the UNFCCC, and it is questionable whether the dispute resolution mechanisms could be used to compel some of the largest emitters, including the United States, that have not yet ratified the traty. Thus, the UNFCCC remains the most likely venue for achieving a global reduction in carbon dioxide emissions.

### 5.2 Non-CO$_2$ Mitigation

The broad pollution controls offered under UNCLOS, which impose management obligations on Parties to limit marine pollution (UNCLOS, 1982a), could be used to encourage states to make greater effort to reduce non-CO$_2$ drivers of ocean acidification. Similarly, UNFSA requires Parties to 'minimize pollution' (UNFSA, 1995) and while no definition of pollution is provided within the agreement text, it could be reasonably interpreted to include pollutants that increase coastal acidification, especially as linkages between increasing acidity and impacts to fisheries become more apparent (Branch et al., 2013). In addition, the Global Programme of Action for the Protection of the Marine Environment from Land-based Activities (GPA), established by the Washington Declaration on Protection of the Marine Environment from Land-Based Activities (UNEP, 1995), provides a forum for limiting nutrient run-off, indeed, the GPA was tasked with working on 'on nutrients, litter and wastewater' and identified a number of land-based sources of pollution including sewage, nutrients, sediment mobilisation, persistent organic pollutants, oils, litter, heavy metals and radioactive substances on which to focus its work (UNEP, 2015). Similarly, the CBD has agreed in Aichi Target number 8, that nutrient pollution





be brought to sustainable levels by 2020 so as not to negatively impact ecosystem function and biodiversity
(CBD, 2010c).
These are all existing measures that can easily be understood to include efforts to reduce the local causes of
ocean acidification. Further, institutions that manage networks of marine protected areas, including, for
example, OSPAR, UNFSA and the Convention on the Conservation of Antarctic Marine Living Resources
(CCAMLR)(1980a), could incorporate the local reduction of acidity into the MPA management as a regular
operating procedure  (Billé et al., 2013). It has been suggested that an implementing agreement under UNCLOS
could provide an avenue for establishing a series of marine protected areas beyond national jurisdictions (CBD,
2006), such areas would be governed by the objectives of UNCLOS, including limiting pollution to the marine
environment. Thus, these MPAs could be established with consideration of ocean acidification and managed
with the intent of responding to it.
**5.3  Adaptation and Protection**
Marine protected areas could also be utilised to enhance resilience and adaptive capacity of ecological and
human communities affected by ocean acidification. The identification and protection of areas that may act as
refugia or hotspots of biodiversity would act to reduce stressors and encourage greater resilience in the face of
ocean acidification. Consideration of ocean acidification could be incorporated into existing strategies and
guidelines for designing and managing MPA networks, such as those designated by the IUCN World
Commission on Protected Areas, which contain guidelines for best practice in regards to climate change (IUCN-
WCPA, 2008) . Similarly, ocean acidification could be incorporated into the General framework for the
establishment of CCAMLR Marine Protected Areas, which already recognises the role of MPAs in contributing
to sustaining ecosystem structure and function and aims to protect areas in order to maintain resilience or the
ability to adapt to the effects of climate change (CCAMLR, 2011). CCAMLR has the ability to designate marine
protected areas that can exclude fishing activities, ship discharges and dumping of wastes, as well as setting
catch limits and designating open and closed seasons for fisheries (CCAMLR, 1980b). All of which could be
useful in protecting species, such as krill, that are likely to be impacted severely by increasing acidity
(Kawaguchi et al., 2011;Kawaguchi et al., 2013) and the Southern Ocean areas that are rapidly acidifying
(McNeil and Matear, 2008). These treaties could be utilised to create networks of protected areas with the
expressed intent of combatting ocean acidification.
The UNFSA and various regional fisheries management organisations (RMFOs) also offer venues for the
consideration of the impact of ocean acidification on fisheries and the management options required to ensure
functional fisheries into the future. These could include the adjusting of take limits and establishing no take
zones to boost resilience in areas most vulnerable to ocean acidification. The CBD could also provide a venue to
host a broader discussion about the integration of ocean acidification into biodiversity adaption and protection
planning. Specifically via the *Climate Change Adaptation Database* that offers guidance on adaptation options
to Parties(CBD, 2017)
**5.4  Restoration**
The CBD also offers an important venue for initiating activities to restore ecosystems degraded by ocean
acidification. Indeed, Article 8(f) of the Convention states that 'each Contracting Party shall […] [r]ehabilitate
and restore degraded ecosystems and promote the recovery of threatened species, inter alia, through the



development and implementation of plans or other management strategies' (CBD, 1992a). Further, Target 14 of
the Aichi Targets requires that 'ecosystems that provide essential services, including services related to water,
and contribute to health, livelihoods and well-being, are restored and safeguarded'(CBD, 2010c). Thus,
restoration of species and ecosystems degraded by ocean acidification fall easily within the CBD mandate. The
RAMSAR Convention on Wetlands (1971) could also offer a venue for restoration activities pertaining to coral
reefs and coastal areas affected by ocean acidification as it contains a very broad definition of wetland that
includes coral reefs and marine waters to a depth of six meters at low tide (RAMSAR, 1971).

### 5.5   Reparations

The UNFCCC COP has initiated efforts to consider ways to address the loss and damages experienced in
developing countries due to climate change (UFCCC, 2017). Interestingly, ocean acidification appears as part of
this discussion; listed as a slow onset event that could result in loss and damage(UNFCCC, 2010). This is the
only mention of ocean acidification in any outcome documents of the COP to date. Here the COP recognised the
need for greater effort to better understand and reduce the loss and damage associated with, among other things,
the impacts of slow onset events, including ocean acidification. In 2013, following two years of deliberations,
the Warsaw International Mechanism for Loss and Damage was established (UFCCC, 2014). It is, as yet,
unclear how the mechanism will progress (Surminski and Lopez, 2014) and how and to what extent ocean
acidification will be factored in. However, this could provide an avenue for addressing issues of reparations
associated with loss and damage resulting from ocean acidification.

### 5.6   Geoengineering

Geoengineering, like ocean acidification, presents numerous governance challenges as it is a cross-sectional
issue that falls under the interest areas of many international institutions while, simultaneously not fitting neatly
within the mandate any one. As yet, there is no clear governance framework applicable to geoengineering.
However, it has been suggested that the use of international environmental impact assessment (EIA)
mechanisms could be an avenue for increasing geoengineering governance (Craik, 2015). This could also offer a
pathway for assessing whether individual geoengineering schemes are likely to be positive, negative or neutral
with regards to ocean acidification.
There are a series of institutions that offer EIA mechanisms that could be utilised for this purpose, including
UNCLOS that imposes an obligation on states to assess the potential effects of planned activities that 'may
cause substantial pollution of or significant and harmful changes to the marine environment' (UNCLOS,
1982b). The CBD also requires states to 'introduce appropriate procedures requiring environmental impacts
assessment of its proposed projects that are likely to have significant adverse effects on biological diversity'
(CBD, 1992b). There is also a clause in the UNFCCC that makes reference to undertaking impact assessments
in order to minimize adverse effects resulting from projects or measures undertaken to mitigate or adapt to
climate change (UNFCCC, 1992b). For regional impacts, the Protocol on Environmental Protection to the
Antarctic Treaty (1991b) offers EIA requirements for all activities occurring within the Antarctic Treaty area
that could have even a 'minor or transitory impact' (ATS, 1991). The Convention on Environmental Impact
Assessment in a Transboundary Context (Espoo Convention)(1991a) may also be applicable as it offers
processes to deal with transboundary impacts, which may occur due to the deployment of geoengineering
mechanisms.





**5.7 Filling the Gaps**
This section has proposed a series of existing instruments that could be utilised in order to enhance the current,
but wanting, international response to ocean acidification. These suggestions include utilising instruments
within institutions that are already responding to ocean acidification, such as the CBD. Opportunities exist
within this institution to expand efforts to mitigate the non-$CO_2$ causes of ocean acidification as well as
adaptation and protection, restoration and geoengineering efforts. Other institutions, such as UNCLOS, that
have previously been inactive with regards to ocean acidification, also offer avenues for increasing the
governance of ocean acidification. UNCLOS offers opportunities for action within efforts to reduce both carbon
dioxide emissions and the non-$CO_2$ causes of rising acidity. UNCLOS can also enhance efforts to adapt and
protect human and ecological systems from acidification, as well as the regulation of geoengineering via
environmental impact assessments.
These previously untapped opportunities offer the potential for addressing each of the six policy domains that
constitute a comprehensive response to ocean acidification. This is visualised in Table 4. The CBD and
UNCLOS are venues that could take on multifaceted responses to ocean acidification. Indeed, these two
institutions could act to guide the wider response across the international community. The UNFCCC is central
to this response with regards to reducing carbon dioxide emissions and is, currently, the only venue discussing
the issue of reparations with relations to loss and damage suffered due to climate change. It is also likely that the
UNFCCC will be abreast of any deployment of geoengineering schemes, and thus, incorporating ocean
acidification into the UNFCCC impact assessment will be an effective way of ensuring impacts with regards to
ocean acidity are considered. While the other institutions listed offer more ancillary avenues to responding to
ocean acidification, they do represent additional opportunities for creating a comprehensive response.
**Table 2: International Institutions Capable of Contributing to a Response to Ocean Acidification**

| INTERVENTION | CBD | UNCLOS | UNFCCC | OSPAR | UNFSA | MARPOL | CCAMLAR | LC&P | GPA | RAMSAR | ATS | Espoo |
|---|---|---|---|---|---|---|---|---|---|---|---|---|
| $CO_2$ Mitigation | | X | X | | X | X | | | | | | |
| Non-$CO_2$ Mitigation | X | X | | X | X | X | X | | X | | | |
| Adaptation and Protection | X | X | | X | X | | X | | | | | |
| Restoration | X | | | | | | | | | X | | |
| Reparations | | | X | | | | | | | | | |
| Geoengineering | X | X | X | X | | | | X | | | X | X |



**6. Conclusion**
This review of activities related to ocean acidification within the UN General Assembly and across the UN
family found that substantive action (rule-making or implementation) to prevent worsening ocean acidification
and to respond to impacts has largely not occurred. Indeed, only three institutions, the London Convention and
Protocol, the OSPAR Convention and the Convention on Biological Diversity, were found to have initiated any
rule making and implementation activities in direct response to ocean acidification. These are useful first steps
in crafting an international response to ocean acidification. However, these policies by themselves are unable to





prevent worsening acidification. This is in large part because these activities focus on the protection of species
and ecosystems and the regulation of geoengineering efforts and do not tackle the root cause of ocean
acidification – rising carbon dioxide emissions.
In addition to these activities there are a number of existing policies, initiated without consideration of ocean
acidification, which are able to help lessen worsening acidification and address its impacts. However, these
activities, in most cases, including carbon dioxide reduction efforts within the UNFCCC, have been found not to
be strong enough to guarantee prevention of ocean acidification in the future. Other mitigation policies,
including those within MARPOL provide positive steps to reduce industry wide emissions of carbon dioxide,
however are not broad enough to capture a large enough segment of emissions so as to prevent future
acidification. In addition, general conservation measures are likely to have a positive effect on ecosystems in the
face of rising acidity. However, without specific intent to address acidification it is possible that such measures
could miss important opportunities with regards to protecting ecological systems. Further, few legally binding
restrictions have been placed on deployment of geoengineering efforts and thus, may be used in the future. It is
important to understand how such efforts could interact with ocean acidification and ensure that the possible
negative impacts are considered prior to their deployment.  Thus, this range of activities were found wanting
with regards to offering a strong, comprehensive response to rising ocean acidity and its impacts.
Therefore, a series of options for enhancing the current international response to ocean acidification utilising, as
yet, untapped instruments was explored. This found, most importantly, that the CBD and UNCLOS could both
take on a substantial amount of work to craft a response to ocean acidification. These two institutions are readily
applicable to responding to ocean acidification due to their interests in conserving biodiversity and protecting
the marine environment. These two institutions could also serve as focal points for and guide a wider
international response to acidification. A number of other institutions were found to have instruments that could
be utilised in responding to rising acidity and its impacts. Including, the UNFCCC that will need to remain the
venue for the mitigation of carbon dioxide with regards to ocean acidification as it is the site of international
efforts to regulate carbon dioxide as a response to climate change.
While this piecemeal approach responding to ocean acidification is perhaps far from ideal it is important to
acknowledge that existing institutions are limited to some extent by their mandates in their capacity to initiate a
holistic response to ocean acidification. For instance, being a regional agreement the OSPAR Convention is
prevented from protecting marine ecosystems globally. The LC&P are limited to governing substances dumped
into the ocean and do not have the capacity to regulate land-based emissions of carbon dioxide. Similarly, the
CBD, while very broad in scope, is probably limited in its ability to regulate carbon dioxide. Despite the lack of
a single institution able to tackle to problem of ocean acidification, this work has found there are a number of
institutions capable of taking on different aspects of a response. Together these efforts, if employed, could cover
all six policy domains that comprise a comprehensive response to acidification. It is unlikely that an existing
institution can or will take on a comprehensive response to ocean acidification. Thus, this piecemeal approach,
while far from ideal, offers a pathway forward that is politically feasible and achievable in the short- to medium-
term – a critical timeframe with regards to this issue.




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





The Author declares no competing interests.

The author would like to acknowledge funding for her research provided by an Australian Post-
Graduate Award.