# Peer review of "An Orphan Problem Looking for Adoption Responding to Ocean Acidification Utilising Existing International Institutions"

_Biogeosciences, 2017_

## Referee Comment (RC1) · Anonymous Referee #1 · 7 Jul 2017

General comments

This paper considers the role of several international institutions (primarily UN bodies) in addressing the problem of ocean acidification. As previously recognised by this author and others, there is no single institution with clear 'ownership' of developing policy responses; the different roles of different bodies are discussed, together with their limitations. The setting-out of such information is of interest, but is not that novel – and some important international policy responses are not covered. The discussion of relevant policy developments in CBD and the London Convention/London Protocol is not up to date. The limitations of the UNFCCC and the Paris Agreement (in not guarantee-

ing that OA will not worsen; an unrealistic expectation) seem to be over-emphasised in relation to the major improvements that they could achieve in comparison to business-as-usual.

Specific comments 1. There would seem three major omissions. Thus there is no mention of: i) UN Sustainable Development Goal 14, and its target 14.3 that requires SDG Parties to "Minimize and address the impacts of ocean acidification, including through enhanced scientific cooperation at all levels"; ii) the role of the Intergovernmental Panel on Climate Change in assessing our scientific understanding of ocean acidification, in particular by WGII in the IPCC 5th Assessment Report, and thereby providing information to UNFCCC and other bodies; and iii) the development of internationally-coordinated ocean acidification monitoring, through the Global Ocean Acidification Observing Network (with several sponsors), in order to develop better understanding of processes, impacts and the potential effectiveness of local mitigation and adaptation measures. All those actions would see important 'policy responses' to ocean acidification.

2. The conclusions of the paper include the statements that "substantive action (rule-making or implementation) to prevent worsening ocean acidification and to respond to impacts has largely not occurred" and that "carbon dioxide reduction efforts within the UNFCC have been found not to be strong enough to guarantee prevention of ocean acidification in the future". The first of those conclusion is questionable, and the second seems politically and environmentally naïve. The only way to 'guarantee' that future ocean acidification will be 'prevented' would be to near-instantly cease all anthropogenic CO2 emissions. The UNFCCC Paris Agreement may not have been specifically designed to combat ocean acidification; nevertheless it arguably represents an extremely ambitious global commitment that, if fully implemented, will reduce future acidification as much as is likely to be socio-economically feasible.

3. There is undoubtedly need to increase the ambition of national contributions to the Paris Agreement, and the issue of ocean acidification is clearly relevant in that context.

There would also seem opportunities for closer working between the UNFCCC and other bodies with regard to ocean acidification. Whilst the desirability of such actions is recognised, it would seem somewhat dismissive to consider them as a 'piecemeal approach'.

4. This reviewer appreciates the rationale for distinguishing policies that explicitly respond to ocean acidification and those that only do so indirectly. Nevertheless there does seem to be overlap, and some repetition, as a result of the paper's structuring. In particular, consideration could be given to combining the information in Tables 2, 3 and 4 (noting that Tables 3 and 4 are currently labelled as Tables 1 and 2 in the Discussion version of the MS). A more comprehensive table (although with somewhat different information) of policy responses to ocean acidification is given as Table 2.1 in CBD Technical Series 75 (Secretariat of the Convention on Biological Diversity, 2014); that table could usefully be updated.

5. The topic of 'Geoengineering' within the paper does not seem to be well addressed. It is initially defined very broadly as "Those policies aimed at manipulating oceanic or atmospheric properties to address ocean acidification, whether they be mitigation, restoration or other type policies". But doesn't that include almost everything? Which particular 'manipulations' are included or excluded? In later text, "the sequestration of $CO_2$ in sub-seabed geological formations" is included; however, that is not widely considered as geoengineering, unless active removal of $CO_2$ from the atmosphere is also involved.

6. The main title of the paper "An orphan problem looking for adoption" is successful in terms of attracting attention; however, the validity of the analogy is questionable. Thus an 'orphan' has lost his/her parents (in what way is that true for ocean acidification?), and the paper would seem to conclude that the 'adoption' by just one body is probably not the best way forward. The secondary title "Responding to ocean acidification utilising existing international structures" is more prosaic, yet also more accurate.

Technical comments

Line 79: "reducing acidity using additives other than iron". Wording seems clumsy; "adding alkaliniity" would be simpler

Lines 82-83: "Non-CO2 mitigation policies": it is not clear whether this is intended to cover measures to reduce emissions of other greenhouse gases (methane, nitrous oxide etc)

Line 93, Table 1: "Interventions for Preventing Worsening OA". It is not clear how the policy domains of adaptation and protection, restoration and reparation 'prevent worsening', since the cause of OA is not addressed.

Line 105: 'negative effects that are already occurring' could also cite i) effects on oyster aquaculture (Barton et al reference given at the end of the para) and ii) the experimental coral growth studies by Allbright et al (2016) Nature 531, 362-365 (doi 10.1038/nature17155)

Line 123: UN Sustainable Development Goals (adopted in September 2015) are surely relevant here, with involvement of UNGA and (for SDG 14) IOC/UNESCO

Lines 141 – 166. This discussion mixes OSPAR's concerns about ocean acidification with CO2 sequestration. The latter is not usually considered to be geoengineering.

Lines 158-159: "carbon dioxide sequestration and storage, which had effectively been banned until this point". Is that correct? Which body had been responsible for such banning?

Lines 175-218: A very detailed account is given of CBD policy discussions on ocean acidification in the period 2008-2012, but nothing since. The CBD's 2014 report on ocean acidification and subsequent COP decisions warrants coverage.

Line 253: change "preventing" to "limiting" (emission reductions can only slow further OA, they won't prevent it. To do that, negative emissions are required)

Line 260: Citation(s) needed to justify the statement that "At 1.5 deg C risk from acidification is likely to be on the verge of high risk". How is 'high risk' defined?

Line 263-264: "It is difficult to conclude that the Paris Agreement, unless implemented in its most stringent form, is strong enough to prevent a worsening of acidification into the future". That statement is technically correct - in that further worsening of acidification (compared to present day conditions) is inevitable. But it is also misleading: the Paris Agreement will, if implemented, greatly reduce the rate of worsening (RCP 2.6 cf RCP 8.5; Gattuso et al, 2015).

Line 267-2268: The focus on MARPOL seems misplaced – if, as stated, it is responsible for only 2.2% of global emissions. Thus CO2 emissions from industry, agriculture, land-use change, aviation and land transport (i.e. the other 97.8%!) are much more important.

Line 269: What has been the effect of the MARPOL (and IMO) measures to increase fuel efficiency in shipping? My understanding is that it has been trivial (less than 1%)

Line 279: Additional references desirable to justify statements on importance of S and N deposition from ships causing local acidification. This effect has been questioned by Hunter et al. (2011)"Impacts of anthropogenic SOx, NOx and NH3 on acidification of coastal waters and shipping lanes." Geophysical Research Letters 38

Lines 281-29: The discussion on ocean fertilization policy discussions by the LC&P and the CBD is not up to date. For update, see Williamson & Bodle (2016) CBD Technical Series 84.

Line 305: What is considered to a 'hotspot' for ocean acidification? Isn't that where protection or other measures might be needed most?

Line 336-7: "leniencies built into the agreement mean that this is not guaranteed". Is it realistic to expect guarantees? The global commitment to keep the temperature increase "well below 2C" is generally considered to be very ambitious, rather than

lenient. It is possible that it may not be fully implemented; nevertheless, it is extremely unlikely (= impossible?) that international agreement could have been reached on anything more demanding.

Lines 520-532: the first seven references do not seem to be in alphabetical order.

---

## Referee Comment (RC2) · D. Herr (Referee) · 7 Aug 2017

I think the topic itself highly interesting and relevant, but would prefer a better structure as well as more forward looking advice/recommendations.

What I have troubles with is separating between chapter 4 and 5. Much of what is discussed in 5, could also be listed under 4. UNCLOS for example, UNFCCC Loss and Damage mechanism. Also, there is no reference to the IPCC reports really, and explanation why the IPCC AR cover OA quite well, but the COP doesn't. Or the new Special Report on Oceans (not sure if it has to do with the timing of the submission). No reference to SDG, SDG 14 in particular. (not sure if it has to do with the timing of

the submission).

Re the CBD WorkPorgramme on Marine and Coastal ecosystem, it surely has more info re non OA action relevant for OA. CCAMLR has some specific Climate resolutions or alike as well. CCAMLR and OSPAR are regional, but there are more regional efforts out there, why only look at them?

Re recommendations: The Rio Conventions report to each other – make OA a topic? An OA Convention is not feasible, what about an global OA Commission under UNC-LOS? Who can ensure the "mainstreaming" of OA in relevant national? Is it a matter of having an NGO OA watchdog looking across the Conventions to ensure action and raise awareness?

I would recommend the paper to go forward with however some revision on structure and explanations of why certain elements are addressed and others aren't.

Please also note the supplement to this comment:
https://www.biogeosciences-discuss.net/bg-2017-230/bg-2017-230-RC2-supplement.pdf

**Supplement:**

[revised manuscript text omitted]

---

## Author Comment (AC1) · 31 Aug 2017

Journal: BG
Title: An Orphan Problem Looking for Adoption: Responding to Ocean Acidification Utilising Existing International Institutions
Author(s): Ellycia R. Harrould-Kolieb
MS No.: bg-2017-230
MS Type: Research article
Special Issue: The Ocean in a High-CO2 World IV

**Author's Response to Referee #1**

I would like to take the opportunity to thank Referee #1 for their very detailed and constructive comments. They have been most helpful in clarifying and strengthening my manuscript. Overall, the comments were on point and have been accepted and will be incorporated into the updated manuscript. Each comment has been laid out in the table below with an author's response.

| Referee Comment | Author Response |
|---|---|
| **Specific Comments** | |
| 1.There would seem three major omissions. All those action would see important 'policy responses' to ocean acidification. i)UN Sustainable Development Goal 14 and its target 14.3 that requires SDG Parties to "Minimize and address the impacts of ocean acidification, including through enhanced scientific cooperation at all levels". ii)The role of the Intergovernmental Panel on Climate Change in assessing our scientific understanding of ocean acidification, in particular by WGII in the IPCC 5th Assessment Report, and thereby providing information to the UNFCCC and other bodies. iii)The development of internationally coordinated ocean acidification monitoring, through the Global Ocean Acidification Observing Network (with several sponsors). | Yes, these are all important policy responses to ocean acidification and are worthy of discussion. However, as will be clarified in a revised manuscript, this paper is interested in the activities taking place within treaty bodies and not within these 'soft-law' instruments. However, they are a very important component of the larger international response to ocean acidification and fodder for future research! |
| 2.The conclusions of the paper include the statements that "substantive action (rule-making or implementation) to prevent worsening ocean acidification and to respond to impacts have largely not occurred" and that "carbon dioxide reduction efforts within the UNFCCC have been found to not be strong enough to guarantee prevention of ocean acidification in the future". The first of those conclusions is questionable, and the second seems politically and | The conclusion will be altered to reflect these comments and those made below. The conclusion will now read: "This review of activities relevant to ocean acidification taking place within treaty bodies has found that they are mostly indirect and are fragmented across a wide swath of regimes, with no central focal point of OA governance." And "Efforts within the UNFCCC to reduce carbon dioxide emissions are critical to the |

| | |
|---|---|
| environmentally naïve. The only way to 'guarantee' that future ocean acidification will be 'prevented' would be to near-instantly cease all anthropogenic CO2 emissions. The UNFCCC Paris Agreement may not have been specifically designed to combat ocean acidification, nevertheless it arguable represents an extremely ambitious global commitment that, if fully implemented, will reduce future acidification as much as is likely to be socio-economically feasible. | lessening of future ocean acidification. However, the current commitments need to be strengthened over time." |
| 3. There is undoubtedly a need to increase the ambition of national contributions to the Paris Agreement, and the issue of ocean acidification is clearly relevant in that context. There would also seem opportunities for closer working between the UNFCCC and other bodies with regard to ocean acidification. Whilst the desirability of such actions is recognised, it would seem somewhat dismissive to consider them as a 'piecemeal approach' | Point taken and the wording will be changed to be more positive and reflect this approach as more of a "patchwork" that has the potential to fill the regulatory landscape |
| 4. This reviewer appreciates the rationale for distinguishing policies that explicitly respond to ocean acidification and those that only do so indirectly. Nevertheless, there does seem to be overlap, and some repetition, as a result of the paper's structuring. In particular, consideration could be given to combining the information in Tables 2, 3 and 4 (noting that Tables 3 and 4 are currently labelled as Tables 1 and 2 in the Discussion version of the MS). A more comprehensive table (although with somewhat different information) of policy responses to ocean acidification is given as Table 2.1 in CBD Technical Series 75 (Secretariat of the Convention on Biological Diversity, 2014); that table could be usefully updated. | Tables 2, 3 and 4 will be combined and structured similarly to the table provided in the CBD Technical Series No. 75. In place of table 2 a timeline of OA activities will be presented. |
| 5. The topic of 'geoengineering' within the paper does not seem to be well addressed. It is initially defined very broadly as "Those policies aimed at manipulating oceanic or atmospheric properties to address ocean acidification, whether they be mitigation, restoration or other types of policies". But doesn't that include almost everything? Which particular 'manipulations' are included or excluded? In later text "the sequestration of CO2 in sub-seabed geological formations" is included; however, that is not widely considered as geoengineering, unless active removal of CO2 from the atmosphere is also involved. | The initial definition will be changed to "policies aimed at a direct manipulation of oceanic or atmospheric properties to counteract climate change and/or ocean acidification and their impacts". This definition is more in line with the definition provided by the CBD Technical Series No. 84.
The Reviewer's point regarding CCS is well taken and the sections where this is addressed will now be titled "CCS and Geoengineering". They remain grouped in the paper as they are often coupled in the discussions taking place within variuos treaty bodies. See for example: CBD Technical Series No. 84, which includes the steps |

| | taken under OSPAR to allow storage of $CO_2$ in sub-seabed geological formations as part of the geoengineering discussion. |
|---|---|
| The main title of the paper "An orphan problem looking for adoption" is successful in terms of attracting attention; however, the validity of the analogy is questionable. Thus, an 'orphan' has lost his/her parents (in what way is that true for ocean acidification?), and the paper would seem to conclude that the 'adoption' by just one body is probably not the best way forward. The secondary title "Responding to ocean acidification utilising existing international institutions" is more prosaic, yet also more accurate. | Title will be changed to: Responding to Ocean Acidification under Existing Multilateral Agreements: Current Responses and Future Possibilities |
| **Technical Comments** | |
| Line 79: "reducing acidity using additives other than iron". Wording seems clumsy; "adding alkalinity" would be simpler | Will change to "adding alkalinity". |
| Lines 82-83: "Non-CO2 mitigation policies": it is not clear whether this is intended to cover measures to reduce emissions of other greenhouse gases (methane, nitrous oxide etc) | Will include: "including non-$CO_2$ greenhouse gases that contribute to OA". |
| Line 93, Table 1: "Interventions for Preventing Worsening OA". It is not clear how the policy domains of adaptation and protection, restoration and reparation 'prevent worsening', since the cause of OA is not addressed | Title of table will be changed to "Interventions for preventing and responding to ocean acidification". |
| Line 105: 'negative effects that are already occurring' could also cite i) effects on oyster aquaculter (Barton et al reference given at the end of the para) and ii) the experimental coral growth studies by Allbright et al (2016) Nature 531, 362-365 (doi 10.1038/nature17155) | Both references will be included. |
| Line 123: UN Sustainable Development Goals 9adopted in September 2015) are surely relvant here, with involvement of UNGA and (for SDG 14) IOC/UNESCO | Yes, a highly relevant development, however beyond the scope of this paper and will be included in future work. |
| Lines 141-166: This discussion mixes OSPAR's concerns about ocean acidification with CO2 sequestration. The latter is not usually considered to be geoengineering | See comment above. Section will now be titled "CCS and Geoengineering" to clarify difference between them. |
| Lines 158-159: "carbon dioxide sequestration and storage, which had effectively been banned until this point". Is that correct? Which body had been responsible for such banning? | The dumping of $CO_2$ into the ocean was banned under the London Protocol as $CO_2$ was not included in Annex 1. Under the LP all substances are banned from being dumped unless included in the Annex. It was the 2006 listing of $CO_2$ in the Annex that then allowed for its disposal in sub-seabed geological formations. This will be clarified in the paper by the inclusion of: "which had effectively been banned until this point due |

| | to the exclusion of $CO_2$ from Annex 1" |
|---|---|
| Lines 175-218: A very detailed account is given of CBD policy discussions on ocean acidification in the period 2008-2012, but nothing since. The CBD's 2014 report on ocean acidification and subsequent COP decisions warrants coverage. | This will be updated to include the 2014 report and most recent COP decision XII/23 |
| Line 253: change "preventing" to "limiting" (emission reduction can only slow further OA, they won't prevent it. To do that, negative emissions is required) | "Preventing" will be changed to "limiting" |
| Line 260: Citation(s) needed to justify the statement that "At 1.5 deg C risk from acidification is likely to be on the verge of high risk". How is 'high risk' defined? | This will be changed to: "Such an agreement would go a long way in reducing the rate of acidification and lessening future impacts. However, there is likely to be a meaningful difference in the impacts experienced at 1.5$^o$C versus 2$^o$C and thus, efforts to maintain the lower level of warming are preferable with respect to ocean acidification (Gattuso et al., 2015;UNFCCC, 2015b)" |
| Line 263-264: "It is difficult to conclude that the Paris Agreement, unless implemented in its most stringent form, is strong enough to prevent a worsening of acidification into the future". That statement is technically correct – in that future worsening of acidification (compared to present day conditions) is inevitable. But it is also misleading: the Paris Agreement will, if implemented, greatly reduce the rate of worsening (RCP2.6 cf RCP 8.5; Gattuso et al, 2015) | This discussion will be reframed to focus on the inability of current NDCs to hold temperature increases to 1.5/2$^o$C and thus there is a need for greater ambition with regards to commitments. Ocean acidification is relevant in this context. New text: "The ambitious Paris Agreement paves the way for large-scale emission reductions, resulting in decarbonisation, thereby preventing future acidification. However, cumulative emissions are still rising and emissions need to rapidly decrease to zero in order to meet the aims of the Paris Agreement (Rogelj et al., 2016). It is generally recognised that current nationally determined contributions (NDCs) do not provide a plausible avenue for meeting the aims of the Paris Agreement and holding temperature increases to well below 2$^o$C (Holz and Ngwadla, 2016;Rogelj et al., 2016). It is suggested that current commitments would result in an overshoot of the target followed by a need for a rapid reduction in emissions including 'negative emissions', or the extraction of $CO_2$ from the atmosphere (Hansen et al., 2017;Rogelj et al., 2016). Such scenarios would result in worsening acidification and irreversible impacts in the near future that could last for tens of thousands of years due to slow ocean processes (Mathesius et al., 2015;UNFCCC, 2015)." |
| Line 267-268: The focus on MARPOL seems misplaced – if, as stated, it is responsible for only 2.2% of global emissions. Thus CO2 emissions | This is the first agreement to regulate emissions from an industry globally – yes, other industries are more significant, but this provides and |

| | |
|---|---|
| from industry, agriculture, land-use change, aviation and land transport (i.e. the other 97.8%!) are much more important. | example of the regulation of a transnational industry. |
| Line 269: What has been the effect of the MARPOL (and IMO) measures to increase fuel efficacy in shipping? | As with the point above, yes, these measures may have resulted in only modest emission reductions, however, it is an important governance model that is not often spoken about. |
| Line 279: Additional references desirable to justify statements on importance of S and N deposition from ships causing local acidification. This effect has been questions by Hunter et al. (2011) "Impacts of anthropogenic SOx, NOx and NH3 on acidification of coastal waters and shipping lanes." Geophysical Research Letters 38 | Will update reference to Hasselov et al. (2013) who finds that: "The calculated near-coastal season acidification of 0.0015-0.002 pH is without a doubt significant: deposition of shipping emissions not only matches the $CO_2$-driven acidification but also reduced the alkalinity of the water." |
| Lines 281-29: The discussion on ocean fertilization policy discussions by the LC&P and the CBD is not up to date. For update, see Williamson & Bodle (2016) CBD Technical Series 84 | This will be updated to include: "Recent developments include the 2013 resolution under the London Protocol (LP.4(8)) that created a new annex, in which prohibited marine geoengineering activities are listed. These activities are prohibited unless they constitute 'legitimate scientific research' and are authorized under a permit (Williamson and Bodle, 2016). To date, the only activity listed under Annex 4 is ocean fertilization." |
| Line 305: What is considered to be a 'hotspot' for ocean acidification? Isn't that where protection or other measures might be needed most? | It has been suggested that maintaining resilience via MPAs is most effective in areas that are least vulnerable to OA, thus, not OA hot spots. To avoid confusion this wording will be updated to: "be specifically located to maintain and support resilience…" |
| Line 336-7: "leniencies built into the agreement mean that this is not guaranteed". Is it realistic to expect guarantees? The global commitment to keep the temperature increase "well below 2C" is generally considered to be very ambitious, rather than lenient. It is possible that it may not be fully implemented; nevertheless, it is extremely unlikely (=impossible?) that international agreement could have been reached on anything more demanding. | As above this discussion will be reframed around current and required NDCs. New text: "cumulative commitments are not currently consistent with the aims of the Paris Agreement, thus, the Paris Agreement institutionalises an iterative process that establishes an expectation of progressively stronger action over time. Parties are expected to take stock of their collective progress and put forward new commitments that increase ambition in future emission reduction plans (Bodansky, 2016). This *ambition mechanism* will bring Parties back together in 2018 for a 'Facilitative Dialogue', which will then be followed by 'global stocktakes' every five years starting in 2023. These assessment and review mechanisms offer an important role in bridging the gap between the aim of the Agreement and national commitments by raising ambition over time (Holz and Ngwadla, 2016). Reports generated by |

| | the various UNFCCC subsidiary bodies and the Intergovernmental Panel on Climate Change (IPCC), including the IPCC report on 1.5$^{o}$C and the UNFCCC periodic review, will help to inform the decision-making process. These offer one avenue for greater consideration of ocean acidification and the risks likely to result from different emission reduction scenarios. |
| --- | --- |
| Lines 520-532: the first seven references do not seem to be in alphabetical order. | These will be amended. |

---

## Author Comment (AC2) · 31 Aug 2017

Journal: BG
Title: An Orphan Problem Looking for Adoption: Responding to Ocean Acidification Utilising Existing International Institutions
Author(s): Ellycia R. Harrould-Kolieb
MS No.: bg-2017-230
MS Type: Research article
Special Issue: The Ocean in a High-CO2 World IV

**Author's Response to Referee #2**

I would like to take the opportunity to thank Dorothee Herr for offering thoughtful and helpful comments on this paper. They have been useful in clarifying the aims of the manuscript and thinking about its structure and purpose. Many of the comments seem to come back to the central question relating to that aims and why certain things been included and others excluded. This, I hope, has been answered in the table below.

| Referee #2 Comments | Author's Response |
|---|---|
| Much of what is discussed in 5, could also be listed under 4. UNCLOS for example, UNFCCC Loss and Damage mechanism. | These two sections are subtly different. The first (section 4) outlines policies and mechanisms that are currently in use and are inadvertently providing a response to ocean acidification. The second (section 5) explores existing mechanisms that are currently not being utilised in a way that is addressing OA (either with purpose or inadvertently). This section, therefore, proposes ways to use existing mechanisms in a response to OA. The UNFCCC Loss and Damage mechanism is included in this section because it is still in the embryonic phase and is yet to be implemented. Thus, it is deemed to be a mechanism that can be utilised in the future to respond to OA. The differences between these two sections will be further clarified in the paper. |
| There is no reference to the IPCC reports and explanation why the IPCC AR cover OA quite well, but the COP doesn't | The aims of the paper have been clarified to include only the activities of treaty bodies and not intergovernmental organisations, including the IPCC. Thus, the activities of the IPCC are beyond the scope of this paper. However, the OA relevant activities occurring within SBSTA will be elaborated upon. |
| No reference to the new Special Report on Oceans (not sure if it has to do with the timing of the submission) | Please see comment above. |
| No reference to SDG, SDG 14 in particular | Please see comment above. SGD 14 along with the Special Report on the |

| | Oceans and the work of the IPCC are all important elements of the broader governance architecture related to ocean acidification. However, as this paper is interested in treaty related activities only these are beyond the scope. These are all fodder for future discussions looking at the broader governance architecture. |
|---|---|
| Re. the CBD Work Programme on Marine and Coastal ecosystems, it surely has more info re. non-OA action relevant to OA. | Yes, the work programme has many activities of relevance to adaptation and resilience etc. These can be more fully considered, however, it is beyond the scope of this paper to consider each in an individually. With that said, the Work Programme will be discussed in more depth in the co-benefits section. |
| CCAMLR has some specific Climate resolutions or alike as well. | Activities of relevance that have occurred under CCAMLR will be included. However, these activities can be categorised as knowledge production and awareness raising, rather than rule-making or implementation. New text: "Ocean acidification has also appeared in the work of the Commission for the Conservation of Antarctic Marine Living Resources (CCAMLR), the decision making body of the Convention on the Conservation of Antarctic Marine Living Resources (CAMLR) since 2009 when in a resolution the Commission acknowledged that increased ocean acidification will result in possible impacts for marine ecosystems.  Ocean acidification was subsequently discussed in the 2010 meeting of the Scientific Committee, at which a number of observer organisations expressed concern over the increase in ocean acidification and its likely impacts.  The Commission further noted the potential importance of ocean acidification and that it is an important issue to consider with regards to the impact of climate change and requested that it be included in a Joint CCAMLR-SCAR Action Group.  The Commission has also discussed the role that of the designation of marine protected areas and marine reserves can play in boosting species' and ecosystem resilience in the face of ocean acidification. Ocean acidification has also received attention under the Scientific Committee where it has been addressed with regards to its likely impacts on krill and a number of reports have been prepared by the Working Group on Ecosystem Monitoring and Management to examine this issue. " |
| CCAMLR and OSPAR are regional, but there are more regional efforts out there, why only look at | These two treaties have been included as they have initiated activities around ocean |

| them? | | acidification, which has not been seen in other regional treaty bodies. However, the Nairobi Convention will also be included as it explicitly mentions OA in its treaty text – the only treaty to do so. |
|---|---|---|
| Suggested Recommendations: | The Rio Conventions report to each other – make OA a topic? | Yes, this is an important suggestion that will be incorporated. |
| | An OA Convention is not feasible, what about a global OA Commission under UNCLOS? | This is an interesting idea. Would this need to be created via a new implementing agreement? What would the role of the Commission be? To coordinate between regimes? Certainly something to think about, however, beyond the scope of this paper as it is looking at already existing mechanisms, this would involve the creation of a new one. |
| | Who can ensure the 'mainstreaming' of OA in relevant national policies? Is it a matter of having an NGO OA watchdog looking across the Conventions to ensure action and raise awareness? | This role could be played by an NGO or perhaps even incorporated into the work of OA-ICC via a database analogous to the Climate Change Adaptation Database held by the CBD. Again, a very interesting discussion, but looking at national activities and that of NGOs/IGOs is beyond the scope of this paper. |